# Segment-specific optogenetic stimulation in *Drosophila melanogaster* with linear arrays of organic light-emitting diodes

Caroline Murawski [1,2], Stefan R. Pulver[3] & Malte C. Gather [1,4✉]

Optogenetics allows light-driven, non-contact control of neural systems, but light delivery remains challenging, in particular when fine spatial control of light is required to achieve local specificity. Here, we employ organic light-emitting diodes (OLEDs) that are micropatterned into linear arrays to obtain precise optogenetic control in *Drosophila melanogaster* larvae expressing the light-gated activator CsChrimson and the inhibitor GtACR2 within their peripheral sensory system. Our method allows confinement of light stimuli to within individual abdominal segments, which facilitates the study of larval behaviour in response to local sensory input. We show controlled triggering of specific crawling modes and find that targeted neurostimulation in abdominal segments switches the direction of crawling. More broadly, our work demonstrates how OLEDs can provide tailored patterns of light for photo-stimulation of neuronal networks, with future implications ranging from mapping neuronal connectivity in cultures to targeted photo-stimulation with pixelated OLED implants in vivo.

[1] Organic Semiconductor Centre and Centre of Biophotonics, SUPA, School of Physics and Astronomy, University of St Andrews, St Andrews KY16 9SS, UK. [2] Kurt-Schwabe-Institut für Mess- und Sensortechnik Meinsberg e.V., Kurt-Schwabe-Str. 4, 04736 Waldheim, Germany. [3] School of Psychology and Neuroscience and Centre of Biophotonics, University of St Andrews, St Mary's Quad, South Street, St Andrews KY16 9JP, UK. [4] Centre for Nanobiophotonics, Department of Chemistry, University of Cologne, Greinstr. 4–6, 50939 Köln, Germany. ✉email: mcg6@st-andrews.ac.uk

Optogenetics uses light to control neural activity with exceptional temporal precision, and since its inception has found widespread applications in cell culture, tissue and animal models (reviewed in refs. [1,2]). The technique is traditionally based on the genetic expression of light-sensitive microbial opsins, i.e., proteins that control the ion flux through the cell membrane in response to light[3]. Through targeted mutations, a large family of channelrhodopsins has been developed, offering activation spectra covering the whole visible spectrum, achieving millisecond temporal response, precise ion selectivity and sensitivity to light intensities from tens of $\mu W\ mm^{-2}$ upwards[2].

Light is typically delivered to targeted cells using inorganic LEDs or lasers[4], which are powerful but rigid, potentially toxic and oftentimes unable to sculpt light with a sufficient degree of spatial resolution. A variety of methods have been used to address this problem, including scanning galvo systems[5], digital micromirror devices[6], projection of LED arrays[7] and implantable LEDs[8–10]. These approaches have been particularly useful when combined with genetic expression systems that can target optogenetic tools to precise subsets of neurons. Despite these advances, options for providing a large number of independent stimulation points remain relatively limited, particularly for use in vivo. The organic light-emitting diode (OLED) technology, which has been successfully commercialised in smartphone and TV displays, has been suggested as an alternative[11,12]. Its inherent ability to form extremely dense pixel arrays with microscopic resolution could enable high-throughput, highly parallelised experiments with thousands or millions of OLED pixels each controlling the activity of individual cells or small clusters of cells. Further advantages of OLEDs over more traditional light sources include the compatibility with mechanically flexible plastic films, low toxicity of involved materials, microsecond response times, generally low cost of manufacturing and simple tuning of the emitted wavelength by the chemical design of the light-emitting molecule used.

We and others have previously shown that OLEDs provide ample brightness to control neurons in culture, in *Drosophila melanogaster* larvae and very recently in vivo[13–16]. In this contribution, we introduce a microstructured OLED array and show that it can be used to test and develop new hypotheses about neural systems by providing highly controlled local photostimulation. We apply the device to study the response of *Drosophila melanogaster* larvae to local excitation and inhibition of the peripheral sensory system. To date, optogenetic control in this animal model was mostly limited to non-structured illumination of whole animals. Locomotion in *Drosophila* larvae is driven by the interaction of central pattern generating (CPG) networks but is also highly dependent on sensory feedback[17,18]. The OLED light source developed here enables the projection of light onto specific areas of interest in a simple and reproducible manner and we use this to control sensory input along the anterior–posterior (A–P) axis of the animal. A long-standing hypothesis in motor system research is that raising and lowering levels of excitability in motor circuits along the A–P axis represents a conserved mechanism for motor programme selection[19,20]. In this study, we focused on segment-scale manipulation of peripheral sensory neuron activity using an optogenetic activator and inhibitor with the aim of controlling excitability along the A–P axis of the larval locomotor system. We find that larval motor responses depend strongly on the location of stimulation and identify a potential decision-making region between segments A2 and A3. We further demonstrate that activation of sensory neurons in individual segments triggers crawling behaviour and that larvae switch between forward and backward crawling depending on the location of the stimulus. Finally, we show that spatiotemporal patterns of sensory cell activation can entrain motor output, depending on the activity state of the network. Our results not only illustrate the substantial value of OLED light sources for the complex interrogation of optogenetic dissections of neural systems, but also provide proof of concept and strong motivation to develop implantable OLED-based light sources for application in larger animal models.

## Results

**Structured OLED illumination for optogenetics.** We selected an OLED device architecture based on a top-emitting design (i.e., light was emitted away from the substrate on which the OLED is deposited). This minimised the distance between the OLED and the targeted cells and thus reduced the angular spread of light. The device design and in particular the pixel size have to be adapted to the larval stage and the cells to be targeted. Here, we aimed at stimulating individual abdominal segments of third instar larvae, which are approximately 400–600 μm wide. We fabricated a linear array of OLED pixels, each 100-μm wide, by photolithographic patterning of the bottom electrode (Fig. 1a, b). Larger OLEDs with a pixel area of $4 \times 4\ mm^2$ were also fabricated to compare segment-specific photostimulation to excitation of whole larvae. To protect the devices from direct contact with water while maintaining high spatial resolution, the devices were encapsulated with 30 μm thin glass substrates. To avoid a strong drop in efficiency when driving the OLEDs at high brightness (socalled "roll-off")[21], our devices are based on a fluorescent blue emitter molecule[22]. The electrical and optical performance of both large and microstructured OLEDs were very similar, but the power density of both was somewhat reduced compared to our

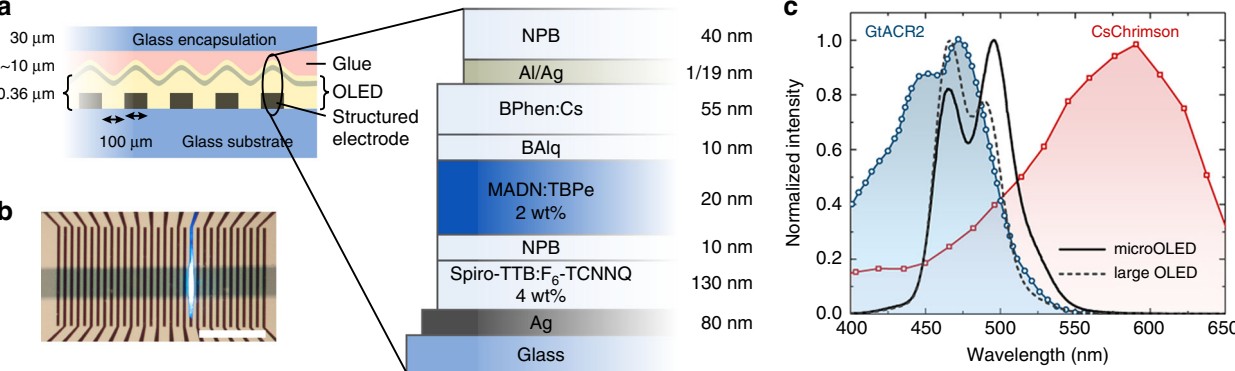

**Fig. 1 Structured OLEDs for optogenetics. a** Device layout and layer structure (not to scale). **b** Photograph of a structured OLED with one active pixel. Scale bar: 2 mm. **c** Electroluminescence spectra of large and microstructured OLEDs and activation spectra of CsChrimson[26] and GtACR2[31].

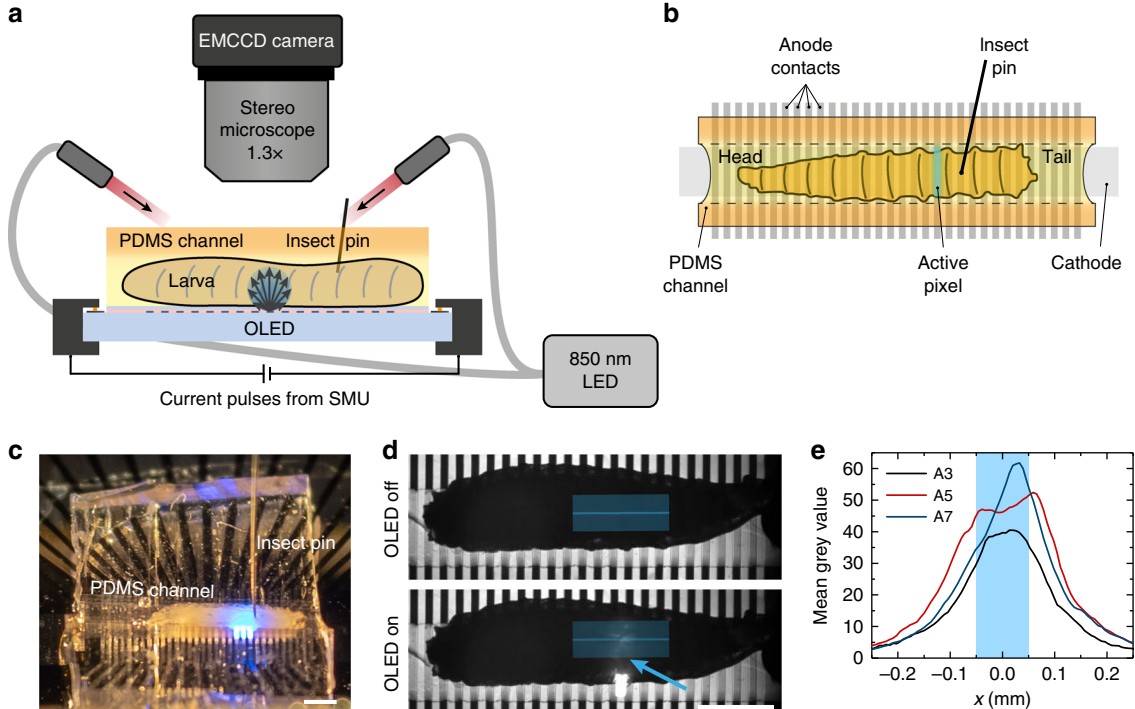

**Fig. 2 Structured light source for segment-specific illumination of *Drosophila* larvae. a** Sketch of the experimental setup (side view) and **b** of the larva mounted on top of the OLED and inside the PDMS channel (top view). **c** Photograph of a larva mounted in a PDMS channel on top of a microstructured OLED. **d** Exemplary images used to estimate the spatial resolution of photostimulation from the profiles of scattered light at the top side of the larva; blue line: extracted emission profile (width: 37 px); the arrow points to the active pixel. **e** Profiles taken during illumination of different abdominal segments. The blue shaded area marks the 100 μm wide OLED pixel. Scale bars in **c** and **d**: 1 mm.

earlier work[13,14,23], which is due to the top-emitting device structure used here (Supplementary Fig. 1). This could potentially be improved in the future by increasing light extraction efficiency through the optimisation of device thickness and the selection of electrodes with higher transparency[24,25].

For excitation, we used CsChrimson, a cation channel with peak sensitivity in the red spectral region[26], which has found widespread application in both flies and mice[27–30]. For inhibition, the anion-conducting GtACR2 was used, which efficiently suppresses neural activity in response to blue light[31], and has been used in previous studies to control *Drosophila* larval locomotion and to study the visual system[32,33]. In contrast to more traditional genetic inhibitors such as tetanus toxin[34], and temperature-sensitive dynamin proteins (shibire[TS])[35], GtACR2 enables fast, reversible inhibition at low light intensities from 2 μW mm$^{-2}$ upwards[32]. Figure 1c compares the activation spectra of both channelrhodopsins to the electroluminescence spectrum of our OLEDs. Since CsChrimson is extremely light-sensitive and also shows a significant response to blue light (sensitivity to 470 nm light from 20 μW mm$^{-2}$)[26], we matched the OLED electroluminescence to the activation spectrum of GtACR2, which cannot be efficiently excited with red light[32]. The contributions from the area and edge emission differ greatly between the microscopic OLED and the large OLED, with edge emission being substantially increased for the microstructured devices (c.f. the spread of light along with the anode contact in Fig. 1b). Additionally, small differences in layer thickness are expected due to the different positioning of the samples in the evaporation chamber. These effects caused small deviations in the overall emission spectrum between both devices (Fig. 1c).

The experimental setup for optogenetic stimulation of restrained *Drosophila* larvae with OLED illumination is shown in Fig. 2a–c. Larvae were mounted and restrained in a

polydimethylsiloxane (PDMS) channel on top of the OLED. Larval behavioural responses were recorded under infrared (850 nm) light illumination, which did not influence larval behaviour in either control or experimental animals (data not shown).

We obtained a conservative estimate for the spatial resolution of the illumination from our microstructured OLEDs by imaging the top side of the larvae under illumination with individual OLED pixels, i.e., collecting OLED illumination after light scattering has taken place across the entire cross-section of the larval body (Fig. 2d). At a 95% confidence interval, the light intensity shows a lateral spread of approximately 160 μm from the pixel centre (Fig. 2e; Gaussian fits give a mean standard deviation of 4σ = 326 ± 53 μm), which compares to a segmental distance of ~400–600 μm in the abdomen of third instar larvae. We expect that the spatial resolution at the ventral side of the animals was even better since scattering and the distance to the light source were strongly reduced.

**Dose response and abdominal sensory stimulation using large OLEDs.** We first tested the optogenetic response of GtACR2- and CsChrimson-expressing larvae on the 4 × 4 mm$^2$ OLEDs. For stimulation, we placed the larvae onto the OLED pixel approximately from segment A1–A2 downwards in order to avoid stimulation of the larval visual system (Fig. 3a). We expressed UAS-GtACR2 and UAS-CsChrimson in all sensory neurons using the 5–40-GAL4 line[17], referred to hereafter as 5–40 > GtACR2 and 5–40 > CsChrimson, respectively. The ">" symbol denotes a driver (here, 5–40-GAL4) expressing a reporter that carries the transgene (here, GtACR2 or CsChrimson). Analysis of the 5–40-GAL4 expression pattern confirmed expression in peripheral sensory neurons and in projections into the larval central nervous system (Supplementary Fig. 2). Third instar 5–40 > CsChrimson

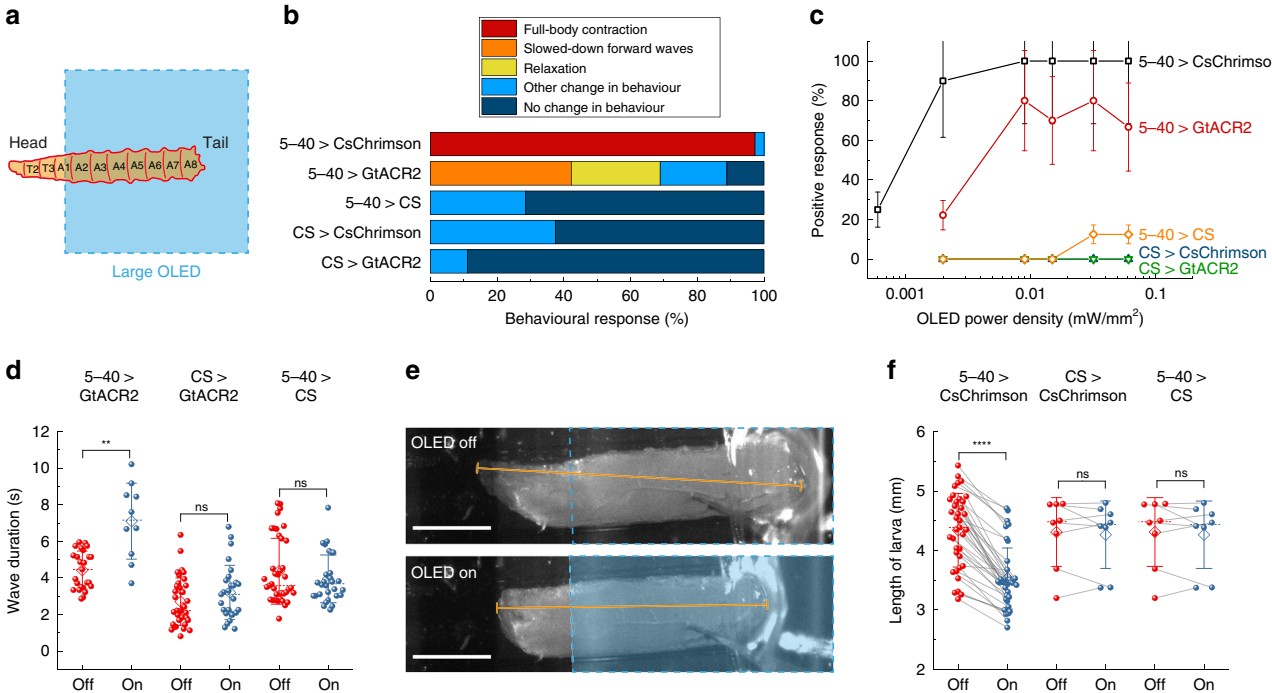

**Fig. 3 Behavioural response upon whole-animal activation/inhibition of sensory neurons using a large OLED. a** Sketch illustrating the placement of larvae on the 4 × 4 mm² OLED. **b** Behavioural response upon photostimulation at 15 µW mm⁻². $n = 17$ larvae for 5–40 > CsChrimson; $n = 20$ for 5–40 > GtACR2; $n \geq 7$ for controls (5–40 > CS, CS > CsChrimson, CS > GtACR2). **c** Dose–response curve, counting either full-body contraction, slowdown of muscle contraction waves or relaxation as a positive response. Larvae expressing 5–40-GAL4 > CsChrimson were stimulated for 3 s with a 12 s break before moving to the next higher intensity; all other larvae were stimulated for 10 s with 10 s breaks (Supplementary Movies 1–3). Data show mean ± SEM; $n \geq 7$ larvae. **d** Wave duration of GtACR2-expressing larvae and controls with and without light application at intensities ranging from 2 to 15 µW mm⁻² showing individual values (spheres), mean (diamond), median (dashed line) and SD (whiskers); each sphere indicates a trial; $n \geq 6$ larvae for each condition. **e, f** Analysis of larval length before and after photostimulation at 15 µW mm⁻² for CsChrimson-expressing larvae and controls. **e** Examples of images used for length measurement. Top: frame recorded immediately before OLED turn-on; bottom: 0.5 s after OLED turn-on. The blue-dashed line outlines the OLED pixel; the orange line indicates the length measurement. Scale bars: 1 mm. **f)** Statistical comparison of larval length; $n = 17$ larvae for 5–40 > CsChrimson, $n \geq 7$ larvae for controls. Two-tailed $t$ test: n.s. not significant ($P > 0.05$), **$P < 0.01$, ****$P < 0.0001$.

larvae were stimulated for 3 s, which was long enough to observe a marked change in behaviour, while 5–40 > GtACR2 and control larvae were stimulated with 10 s long light pulses. Figure 3b, c shows behavioural responses at 15 µW mm⁻² and the dose–response curve, respectively.

GtACR2-expressing larvae responded either with a full-body muscle relaxation that led to immobilisation or with significantly slowed-down muscle contraction waves when exposed to more than 8 µW mm⁻² of OLED illumination (Fig. 3b, c and Supplementary Movie 1). These observations are in line with previous studies, where sensory neurons were inhibited using shibire[17,18] or NpHR[36], and which showed a slowdown of muscle contraction waves and eventual immobilisation of larvae at prolonged inhibition. Furthermore, we observed that the mouth hooks were still moving even if larvae were immobilised, again in agreement with the previous studies[18]. Overall, only 65–80% of GtACR2-expressing larvae showed a marked reaction to light (i.e., slowdown of muscle contraction waves or relaxation). Nevertheless, muscle contraction waves slowed down significantly upon inhibition; the average wave duration increased from 4.4 to 7.1 s (Fig. 3d).

CsChrimson-expressing larvae reacted to light with intensities from 2 µW mm⁻² upwards and showed a full-body muscle contraction that was typically held for the entire time that the light source was on (Supplementary Movie 2). A similar "accordion-like" muscle contraction has also been observed earlier for sensory stimulation with the 5–40-GAL4 driver[37] and is probably due to activation of class II and III multidendritic

(md) neurons[38]. To quantify this behaviour, we measured the length of the larvae and observed a significant shortening upon light turn-on (Fig. 3e, f).

In contrast to the intensity thresholds for activation in literature[26,32], in our experiments, CsChrimson-expressing larvae generally showed a lower response threshold than GtACR2-expressing larvae. Note that the intensity requirements generally depend on the targeted cells, developmental stage, the concentration of supplemented all-*trans* retinal (ATR), the spectrum of the used light source and stimulation time. Hence, a direct comparison to the literature is not possible.

Heterozygous controls without CsChrimson or GtACR2 expression (i.e., crosses of 5–40-GAL4, CsChrimson and GtACR2 to wild-type Canton Special (CS) flies) showed neither relaxation nor slowdown of muscle contraction waves (Fig. 3b, d) nor full-body contraction (Fig. 3b, f). However, especially at high brightness, we observed that control larvae responded to the visual input with quick head and tail movements and increased crawling speed (Supplementary Figs. 3, 4 and Supplementary Movie 3). While OLEDs can heat up when driven at high currents for extended times, a worst-case estimate (no heat dissipation, all electrical power converted to heat) shows that surface temperatures in our experiment will not have risen by more than 3.4 °C; previous studies have measured a 1.1 °C increase in temperature for similar light intensity levels but inferior OLED performance[15]. Thus, it is highly unlikely that heating caused the observed response in control larvae. Instead, the response is likely due to

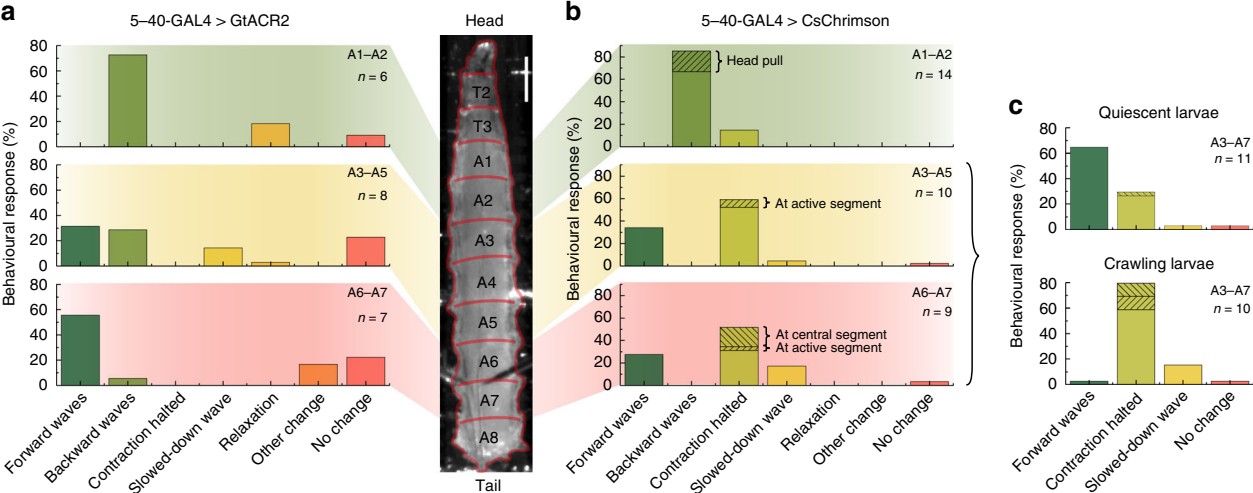

**Fig. 4 Behavioural response upon local inhibition/activation of sensory neurons using a microstructured OLED. a** 5–40-GAL4 > GtACR2 larvae illuminated at segments A1-A2 (top), A3-A5 (centre) and A6-A7 (bottom). Illumination pulses lasted 10 s and used three neighbouring OLEDs. *n*, number of larvae tested. **b** 5–40-GAL4 > CsChrimson larvae illuminated at segments A1-A2 (top), A3-A5 (centre) and A6-A7 (bottom). Illumination pulses lasted 3 s and used a single OLED pixel. **c** Behaviour of 5–40-GAL4 > CsChrimson larvae upon illumination of segments A3-A7, discriminating between larvae that were quiescent before stimulation (top) and larvae that were previously crawling (bottom). Scale bar: 500 μm. Light intensity: 15 μW mm$^{-2}$.

activation of class IV body wall sensory neurons, which mediate a light avoidance behaviour upon exposure to blue light[39].

**Segment-specific sensory stimulation with microstructured OLEDs.** Having confirmed the response of larvae to optogenetic stimulation of the entire abdominal sensory system with large OLEDs, we next used our microstructured OLEDs to look at the larval response to local inhibition and activation of sensory neurons in individual segments along the A–P axis of the animal. Figure 4a summarises the behavioural response of GtACR2-expressing larvae for illumination of three different abdominal regions: Inhibition of sensory neurons in A1–A2 mainly evoked backward waves, with a few larvae showing relaxation. Posterior inhibition in segments A6–A7 instead predominantly evoked forward waves and inhibition of intermediate segments (A3–A5) caused a mixed response of forward and backward waves. Overall, the local inhibition of sensory neurons caused a significantly different behaviour from the whole-animal inhibition, with only a few larvae showing relaxation or slowdown of muscle contraction waves. Notably, the observed behaviour was absent in controls (Supplementary Fig. 5) and the effects occurred during rather than after excitation, so we can exclude the possibility of the observed behaviours being a result of rebound firing in sensory neurons after light turn-off.

For CsChrimson-expressing larvae, we made several interesting observations upon local activation of sensory neurons in single segments (Fig. 4b): anterior stimulation (A1–A2) predominantly caused backward crawling or a head pull—a response that has previously also been observed for mechanosensory stimulation at thoracic segments[40]. Stimulation in more intermediate or posterior segments (A3–A7) showed two different behaviours—either forward waves were evoked or the muscle contraction wave was temporarily halted at the illuminated segment and subsequently resumed after removal of the optical stimulus (Supplementary Movie 4). Here, no significant difference between activation of intermediate and posterior segments was found. However, for posterior stimulation, muscle contraction waves were more often halted in central segments (A3–A5) instead of being paused at the illuminated segment. Taking into account the behaviour of the larva before stimulation, it becomes evident that

quiescent larvae mainly responded with forward crawls, while larvae that were already crawling halted the muscle contraction wave upon optogenetic stimulation (Fig. 4c).

Interestingly, evoking forward waves for posterior stimulation and backward waves for anterior stimulation are observed for both activation and inhibition of sensory neurons. Furthermore, both GtACR2- and CsChrimson-expressing larvae were slightly more responsive for anterior stimulation than for posterior stimulation (as seen by the increased fraction of larvae with no change in behaviour for the latter). This suggests a gradient in excitability along the anterior to posterior axis, with higher excitability in anterior segments.

In the next set of experiments, we explored the optical halting of contraction waves observed in CsChrimson-expressing larvae in more detail. Figure 5a shows individual frames from a representative time-lapse movie (Supplementary Movie 4), where segment A5 was stimulated for 3 s. At the start of the stimulation (0 s), a muscle contraction wave had just reached segment A3 and continued to progress through the animal. The subsequent forward wave, however, stopped when reaching the photostimulated A5 segment (2 s). After photostimulation was stopped, the contraction wave continued to propagate from the halted segment to A1 (4.5 s). To study the temporary optical halting of muscle contraction waves in more detail, we tracked the spatiotemporal location of the muscle contraction wave over time for different segmental distances, $n_{seg}$, between the segment contracted at the time of light turn-on and the stimulated segment (Fig. 5b, c). Independent of the actual point of stimulation, forward wave activity generally continued until the contracting segment reached the light source.

We further looked at the time response of this behaviour. The mean duration of muscle contraction waves increased from 4.2 to 6.4 s under illumination and it took on average 0.85 s after light turn-off until larvae resumed crawling (Fig. 5d, e). The wave duration increased by up to 3 s, with this upper limit given by the 3 s long illumination period (Fig. 5f). Where the increase in wave duration was less, this was either due to the contracted segment lying far away from the stimulated segment (e.g., trace v in Fig. 5b), or due to extremely slow wave progression (trace ii). Finally, we calculated the time that elapsed between turning the OLED on and stopping of the muscle contraction wave ($t_{stop}$) and normalised this

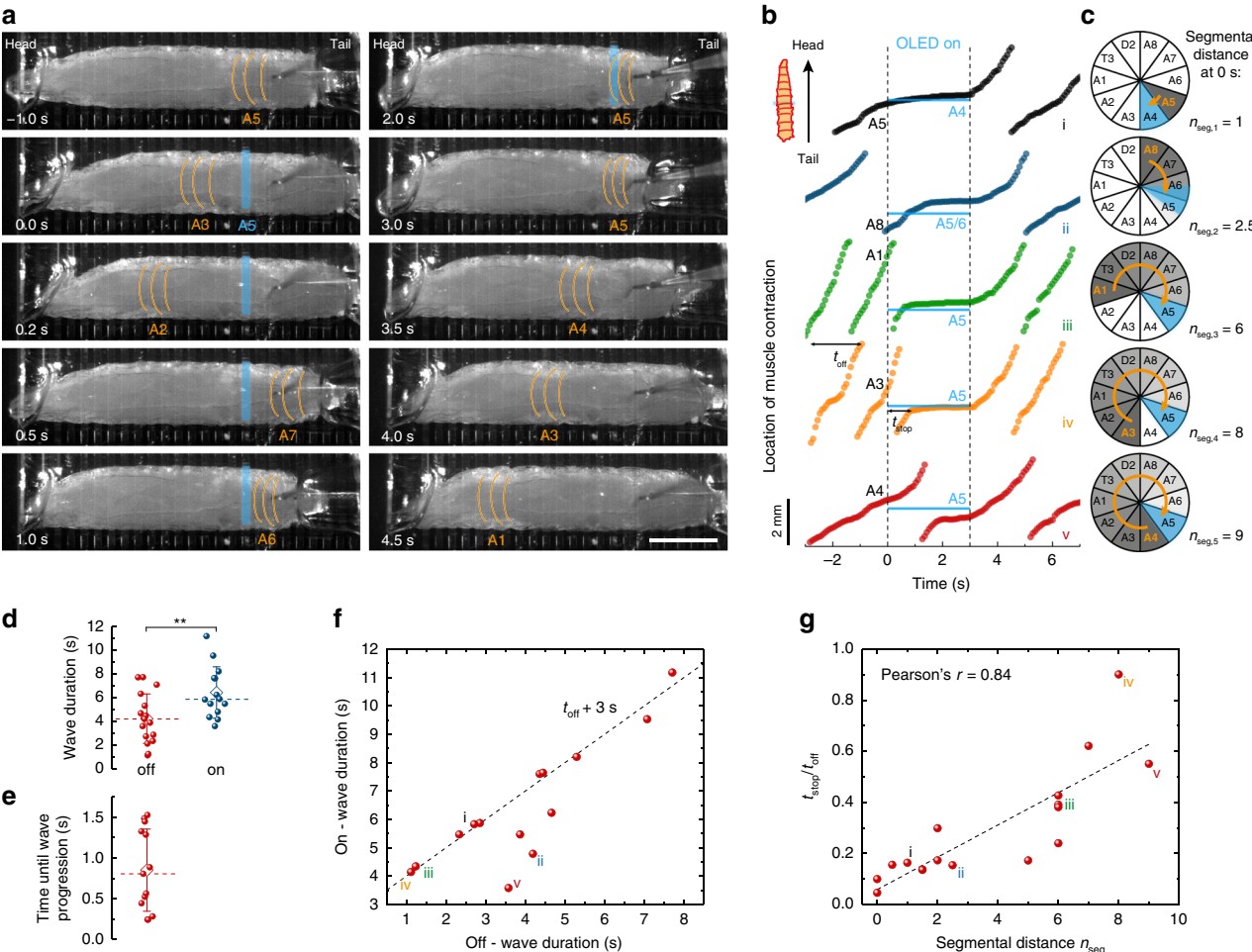

**Fig. 5 Local optical halting of muscle contraction waves upon sensory neuron activation in CsChrimson-expressing larvae.** Larvae were stimulated at posterior segments A3–A7 with one OLED pixel at a light intensity of $15\,\mu W\,mm^{-2}$ for 3 s. **a** Representative time lapse indicating the active larval segment with orange lines in each frame. Active OLED pixel located at A5 as indicated in blue. Scale bar: 1 mm. See Supplementary Movie 4 for complete time lapse and trace iv (orange) in **b** for further analysis. **b** Location of forward muscle contraction waves over time. The location of the active OLED and of the illuminated larval segment is indicated in blue. The segment that was contracted at the time of light turn-on (at 0 s) is indicated in black. Data shown were collected from four different larvae. **c** Polar graphs illustrate the progress of muscle contraction waves between the time of OLED turn-on and the wave coming to a local halt. The contracted segment at OLED turn-on marked in orange; illuminated segment marked in blue. Segmental distance, $n_{seg}$, represents the number of segments between the segment contracted at the time of light turn-on and the stimulated segment. **d** Statistical comparison of wave duration for OLED off and on. During on-time, wave durations were significantly longer ($P = 0.008$, two-tailed $t$ test) than during off-time. Whiskers: SD; diamond: mean; dashed line: median. **e** Statistical analysis of time elapsed between light turn-off and the progression of the muscle contraction wave. **f** Relation between durations of contraction wave under illumination, $t_{on}$, and in the dark, $t_{off}$. Dashed line: $t_{on} = t_{off} + 3$ s. **g** Time that elapsed between light turn-on and pausing of muscle contraction wave, $t_{stop}$, over $t_{off}$ as a function of $n_{seg}$ ($n_{seg}$ as defined in **c**). Roman numbers i–v in **f** and **g** indicate the corresponding graphs in **b**. $n = 6$ larvae with ≥13 stimulations for **d**–**g**.

value by dividing with the wave duration when the light was off ($t_{off}$). Figure 5g shows $t_{stop}/t_{off}$ as a function of the segmental distance $n_{seg}$. A linear relationship with a slope of $0.06 \pm 0.01$ was obtained, which further supports that ongoing muscle contraction waves progressed until they reached the point of stimulation. Note that a slope of less than 0.1 indicates that muscle contraction waves accelerate after stimulation (one full muscle contraction wave includes ten segments, i.e., $n_{seg} = 10$ (c.f. Figure 5c), and $t_{stop}/t_{off} = 1$ for a complete, undisturbed muscle contraction wave). Thus, larvae integrated the altered sensory input but, instead of an immediate reaction, first completed the current muscle contraction wave before switching to another behaviour.

**Switching locomotion direction with targeted stimulation.**
Next, we used our microstructured OLED arrays to trigger a

specific crawling mode in larvae and then switch the direction of crawling by targeted sensory stimulation. This enabled us to test whether the behaviour upon sensory input depends on the current crawling mode and if the susceptibility to sensory input depends on the current phase within a contraction wave. As we saw earlier, stimulation of CsChrimson-expressing larvae in segments A1–A2 caused larvae to crawl backward while central and posterior stimulation in segments A3–A7 evoked forward waves. Typically, larvae maintained optogenetically induced crawling behaviour for some time, with around 1–9 subsequent forward crawls in the case of posterior stimulation, and 0–4 backward crawls upon anterior stimulation (Fig. 6a, b).

Figure 6c shows the location of a muscle contraction wave over time for a representative stimulation sequence (Supplementary Movie 5). Every 10 s, an OLED pixel was turned on for 3 s, alternating between stimulation of anterior or posterior segments.

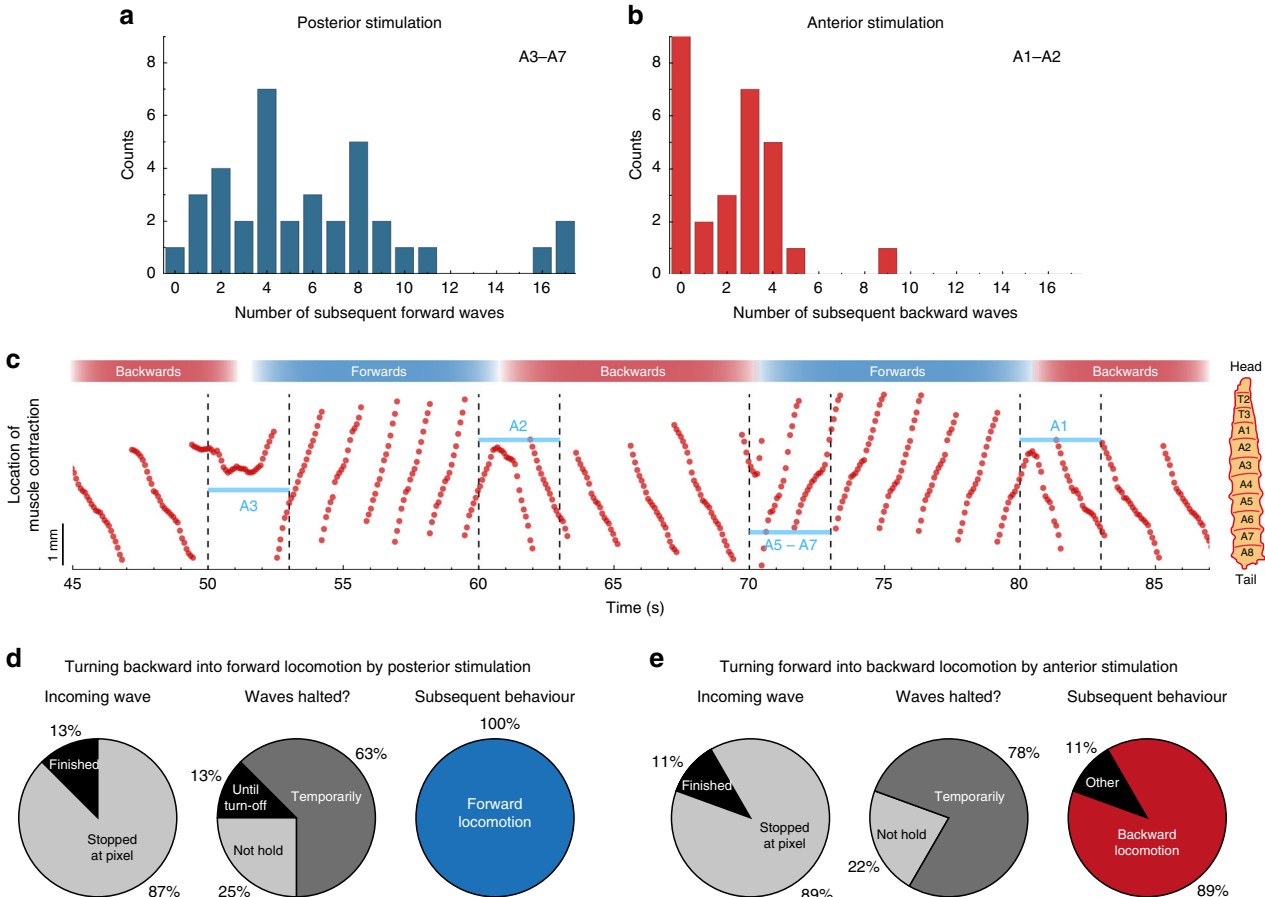

**Fig. 6 Switching locomotion direction in 5–40-GAL4 > CsChrimson larvae by targeted local stimulation. a** Histogram of the number of subsequent forward crawls within a detection window of 30 s after stimulation in segments A3–A7 for 3 s. $n = 10$ larvae, 36 stimulations. **b** Same as **a** for backward waves after anterior stimulation. $n = 14$ larvae, 28 stimulations. **c** Representative trace of the location of a muscle contraction wave over time (Supplementary Movie 5). Every 10 s, the larva was stimulated for 3 s, alternating between stimulation in more anterior or more posterior regions. The illuminated segment is indicated in blue. **d**, **e** Statistics of switching behaviour for **d** posterior stimulation ($n = 3$ larvae, eight stimulations) and **e** anterior stimulation ($n = 5$ larvae, nine stimulations). Light intensity: 15 μW mm$^{-2}$.

The slope of the graph switched accordingly, between negative for backward waves (anterior stimulation) and positive for forward waves (posterior stimulation). Interestingly, muscle contraction waves were not paused during the entire illumination time but instead frequently appeared to "bounce off" the photostimulated muscle segment and to switch direction.

To test whether this is a more general behaviour, we statistically analysed the behavioural changes from multiple larvae and stimulations (Fig. 6d, e). Upon optical stimulation, incoming muscle contraction waves usually stopped when reaching the illuminated segment (in around 90% of the stimulations). Subsequently, muscle contraction was typically only temporarily paused (60–80%) before the larva continued to crawl in the opposite direction (90–100%). The temporary halt observed here is contrary to our earlier observations, where muscle contraction waves were stopped for the entire duration of the stimulus (Figs. 4c and 5). Note that here we applied a posterior stimulus to backward-crawling larvae and an anterior stimulus to forward-crawling larvae, thus switching crawling direction, while in our earlier experiments, we applied a posterior stimulus to forward-crawling larvae (leading to interrupted contraction waves but without a change in crawling direction). Hence, we conclude that the way in which sensory input is processed depends on the current behavioural mode.

**Application of wave-like stimulation**. The capability of our densely structured OLEDs to provide spatiotemporal stimulation is ideally suited to deliver complex light patterns. The muscle contraction waves that underlie forward crawling in *Drosophila* larvae are generated in the first instance by CPG networks in the larval central nervous system; however, the extent to which patterns of sensory neuron activity can entrain motor output has not been thoroughly examined[41,42]. In the next set of experiments, we therefore investigated whether it is possible to influence and/or override the activity of CPG networks by delivering a wave-like sensory stimulation.

The typical wave duration of 5–40-GAL4 > CsChrimson larvae in our study was around 4 s (Fig. 5d). To mimic this crawling speed, we delivered a light wave across the abdomen from posterior to anterior, subsequently turning on six OLED pixels (each 100 μm in width) that were spaced at a pitch of 400 μm for 0.5 s each. Thus, the light wave travelled a total distance of 2 mm over the course of 3 s. Figure 7a shows representative time-lapse images of larval response to the wave-like illumination sequence (Supplementary Movie 6). Larvae that were quiescent before illumination typically initiated a muscle contraction wave that followed the wave-like stimulation, as evidenced by tracking the location of the muscle contraction waves of several larvae (Fig. 7b). However, crawling patterns appeared somewhat less natural, showing tight contraction of illuminated segments and

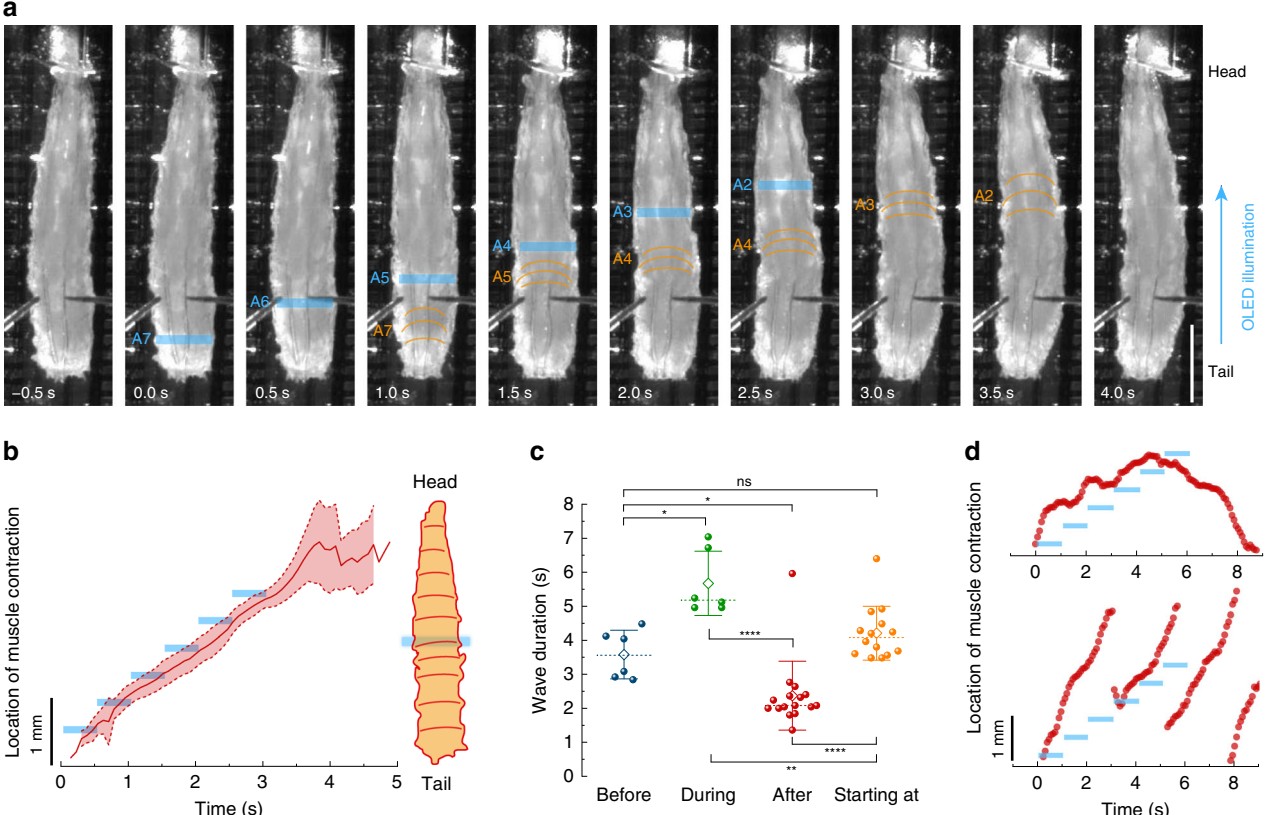

**Fig. 7 Response of 5–40-GAL4 > CsChrimson larvae to wave-like optical stimulation from posterior to anterior. a** Representative time-lapse images. The contracted larval segment is indicated by orange lines in each frame. Active OLED pixels are indicated in blue. Scale bar: 1 mm. See Supplementary Movie 6 for complete time lapse. **b** Overlay traces showing mean ± SD of the location of muscle contraction waves over time. The timing and location of the active OLED pixel are indicated by blue bars. $n = 3$ larvae, nine stimulations. **c** Wave durations grouped by start and end time of the wave relative to the optical stimulation. Before: wave ended before optical illumination. During: wave started before illumination and ended after illumination ceased. After: wave started after illumination. Starting at: wave started during illumination. $n = 4$ larvae. Two-tailed $t$ test: n.s. not significant ($P > 0.05$), *$P < 0.05$, **$P < 0.01$, ****$P < 0.0001$. Whiskers: SD; diamond: mean; dashed line: median. **d** Two representative traces showing the location of muscle contraction in response to slowed-down wave-like stimulation. Light intensity: 30 µW mm$^{-2}$.

barely any contraction of thoracic segments (see Fig. 7a and Supplementary Movie 6).

Figure 7c compares the duration of muscle contraction waves before illumination, after illumination, for waves that started during illumination and for waves that started before illumination and ended after illumination. Before illumination, wave durations were comparable to the previous results shown in Fig. 5d. Waves that were initiated by the light wave showed similar durations as before illumination, in line with the 3 s duration of the light wave. After illumination, waves significantly accelerated. Due to sensory activation, waves that started before illumination were significantly slowed down relative to all other conditions.

We also investigated the response of larvae to slower light waves (total wave duration of 6 s, with 6 steps of 1 s each). Two distinctive behavioural patterns were observed: if larvae were quiescent prior to stimulation, then they could entrain with the slow stimulation speed initially; however, the induced forward wave was converted into a backward wave when the illumination reached anterior abdominal segments (Fig. 7d, top and Supplementary Movie 6). Larvae that were already crawling prior to the stimulation could not adapt to the speed of the slow illumination and instead progressed with fast forward waves (Fig. 7d, bottom and Supplementary Movie 6). This suggests that neural circuits within segments are tuned to receive sensory input within a specific time window during motor waves. Outside this

temporal window, the network is less able to entrain with the speed of sensory stimulation.

## Discussion

Sensory feedback plays an important role in *Drosophila* larval locomotion. Using both macroscopic and microscopic OLED pixels, we activated and inhibited sensory neurons in the ventral nervous system of the entire larvae and within individual larval segments. We observed that segment-specific activation and inhibition of sensory neurons initiated waves. Circuit mechanisms enabling triggering of waves after both activation and inhibition of sensory neurons remain unclear. However, this observation leads to a testable hypothesis for future work, namely that the larval locomotor system is based on sensory neuron activity remaining within a certain activity range instead of simple ON- and OFF-states[37]. Additionally, the dense patterning of our OLEDs allowed us to deliver wave-like optical stimulation to sensory cells, with which we were able to entrain locomotor waves. While additional investigations are required to reveal the underlying mechanisms, our observations suggest the existence of a specific time window during motor waves during which motor circuits can be entrained by sensory input. We also observed a location-specific sensory response with stimulation in segments A1–A2 leading to backward crawling and stimulation in A3–A7

triggering forward crawling. From this, we conclude that signals received from sensory neurons located anterior or posterior of the region around A2/A3 are processed in different ways as has been demonstrated in previous work[43]. After optically triggering a specific crawling mode, larvae continued in this mode for several seconds, which suggests the presence of short-term memory mechanisms downstream of sensory neurons that enable initiation of multiple motor events. Overall, our work suggests that control of excitability along the A–P axis by sensory neurons in this system enables state-dependent initiation of motor activity and that motor networks within segments are tuned to be receptive to sensory input within defined windows relative to motor activity.

Using OLED technology allowed us to deliver light in a more spatially controlled fashion and thus to specific regions of *Drosophila* larvae. This has offered useful insight into the functioning of the *Drosophila* neural system. The technique could be readily adapted to explore how sensory input is integrated in a behavioural state-dependent manner. Specifically, OLED technology could be used to trigger and maintain particular behavioural states (e.g., backward crawling) during experiments. In the future, spatially controlled illumination with OLEDs can be combined with GAL4 drivers that restrict expression of channelrhodopsins to specific classes of sensory neurons to further disentangle the role of sensory feedback for larval locomotion. We expect that even smaller subsets of neurons in *Drosophila* larvae may be targeted by using red light to reduce scattering within larvae and by shaping the OLED emission more into a forward direction, e.g., by increasing the strength of the optical microcavity[44]. Furthermore, flexible OLEDs could be shaped to form a tube through which larvae could crawl and which would give better optical access to neurons along the dorsal–ventral axis.

Our results provide a representative example of using OLEDs for optogenetic stimulation with greatly improved spatial targeting. In contrast to other light sources, OLEDs can be produced in nearly any shape or size. This could enable µm-scale illumination that would be particularly interesting for studying signal progression in interconnected neural networks in vitro[45]. On the other extreme, centimetre-sized light sources would be beneficial for delivering homogeneous illumination to entire organs in vivo, e.g., for optogenetic control of bladder function[46]. OLEDs can also be stacked on top of each other[47], to enable co-localised multi-colour optogenetic excitation. Further, they can be made transparent[48], which allows for simultaneous imaging through the light source. In combination with recent and ongoing developments of devices with improved mechanical flexibility[49], we expect that OLEDs can be integrated onto implants and thus will also find application in freely moving animals, where they may be used to dynamically control neural networks at currently unprecedented spatial resolution.

## Methods

**OLED fabrication and characterisation.** OLEDs were fabricated on 1.1 mm thick cleaned glass substrates (Eagle XG, Dow Corning Inc.) in a vacuum chamber (Evovac, Angstrom Engineering) at a base pressure of $10^{-7}$ mbar. The following layer structure was used: 80 nm Ag anode, 130 nm 2,2′,7,7′-tetrakis(N,N′-di-p-methylphenylamino)-9,9′-spirobifluorene (Spiro-TTB) *p*-doped with 2,2′-(perfluoronaphthalene-2,6-diylidene)dimalononitrile (F6-TCNNQ) at 4 wt% as hole transport layer, 10 nm N,N′-di(naphthalene-1-yl)-N,N′-diphenylbenzidine (NPB) electron blocking layer, 20 nm 2-methyl-9,10-bis(naphthalen-2-yl)anthracene (MADN) doped with 2,5,8,11-tetra-tert-butylperylene (TBPe) at 2 wt% as emission layer, 10 nm bis-(2-methyl-8-chinolinolato)-(4-phenyl-phenolato)-aluminium(III) (BAlq) hole blocking layer, 55 nm 4,7-diphenyl-1,10-phenanthroline (BPhen) *n*-doped with Cs at 2 wt% as electron transport layer, 1 nm Al and 19 nm Ag cathode and 40 nm NPB capping layer. All layers were evaporated through shadow masks without breaking the vacuum. The thickness was controlled in situ using quartz crystal monitors. For microOLED devices, the bottom Ag anode was structured via photolithography using a spin-coated bilayer photoresist of 660 nm LOR7B (Microchem, 3000 rpm, baking at 180 °C for 10 min) and 2 µm S1818

(Microchem, 5000 rpm, baking at 100 °C for 1 min). Patterns were developed for 40–50 s in MF319 (Microchem), the Ag anode was evaporated and the photoresist was then lifted off in acetone (≈ 5 min) and MF319 (10 s). The samples were rinsed in deionised water and isopropanol and heated out at 120 °C in a nitrogen-filled glovebox prior to OLED fabrication. After layer deposition, OLEDs were encapsulated in the glovebox with 30 µm thin glass substrates (Schott) carefully pressed onto the samples using a UV-curable epoxy (NOA68, Norland Products). Microstructured OLEDs measured 100 µm × 1 mm (area of 0.104 mm²), with 100 µm gaps between adjacent pixels, while large OLEDs had an active area of 16.9 mm².

OLED characteristics were measured with a source-measure-unit (Keithley 2400), a calibrated Si photodiode and a calibrated spectrograph (Oriel MS125) coupled to a CCD camera (Andor DV420-BU). The power density was calculated assuming Lambertian emission.

**_Drosophila_ strains and culturing.** The following fly strains were used: Canton Special (CS) wild-type, 5–40-GAL4[17], UAS-CsChrimson inserted in the AttP2 landing site[26] and UAS-GtACR2-EYFP inserted in the VK00005 landing site[32]. Fly crosses were cultured in the dark at 25 °C on conventional cornmeal-agar medium supplemented with 0.5 mM ATR. As a control, 5–40-GAL4-, CsChrimson- and GtACR2-flies were crossed to CS flies and raised as well on ATR-supplemented food. All fly lines are freely available from the authors upon reasonable request.

**Fabrication of PDMS channels.** For fabrication of PDMS channels[50], a casting mold was created via 3D printing (Makerbot Replicator 2) with a channel width of 1.16 ± 0.04 mm and a height of 0.63 ± 0.10 mm. PDMS (Sylgard 184, Dow Corning) was prepared as recommended, poured into the mold and cured at room temperature until hardened. Subsequently, a square-shaped piece of ~8 mm width was cut out and placed on top of the OLED. The channel was filled with water, which allowed clear imaging of the larvae and good adherence of the PDMS channel to the OLED.

**Anatomical analysis of expression patterns.** Third instar 5–40 > CsChrimson animals were filleted and pinned in a sylgard lined dish using insect pins. Dissected preparations were fixed in 4% paraformaldehyde, then washed three to four times in 0.1 M phosphate buffer. Preparations were mounted in Vectashield® (Vector Labs, Burlingame, CAe) and imaged with a Zeiss ApoTome.2 imaging system (Carl Zeiss Microscopy, Jena, Germany).

**Optogenetic imaging and data analysis.** All optogenetic measurements were recorded underneath a stereomicroscope (Nikon SMZ25) with an EMCCD camera (Andor Luca). Images for Fig. 2d, e were taken with an upright microscope (Nikon Eclipse Ni-U) and sCMOS camera (Andor Neo). For behavioural studies on large OLEDs, an additional 500 nm long-pass filter was mounted in front of the camera to avoid overexposure. Larvae were illuminated with a custom-made infrared LED light source (Thorlabs M850L3; 850 nm peak, 30 nm FWHM) projected onto the sample via a dichroic mirror (805 nm cutoff wavelength) and a Y-branched fibre bundle, causing a light intensity of ~0.24 mW mm⁻² at the sample. A custom Python programme was used to trigger the camera and supply constant current to the OLED via an SMU (Keithley 2450). For experiments with wave-like illumination, OLEDs were addressed and driven by an Arduino Mega. The voltage output of the Arduino was converted to constant currents of 0.226 mA per pixel by constant current drivers (LM334Z). Movies were recorded as kinetic series in Andor Solis software at 25 Hz/16 bit and exported as tiff images. The exposure time was kept to 10 ms or less to avoid motion blur. Exposure time and camera gain were adjusted according to the brightness of the OLED and illuminating infrared LED.

For optogenetics experiments, feeding third instar larvae were taken out of the vials in dim light and gently washed in water. Then, larvae were slowly pushed into the water-filled PDMS channel from one side, while water was removed from the other side, to suck the larvae into the channel. Subsequently, larvae were pinned down dorsal side up with an insect pin positioned between the trachea approximately in segment A5–A6. After fixing larvae in the PDMS channel, a slight adjustment of the larval position with respect to the OLED pixel was possible by carefully sliding the PDMS sheet on the glass surface of the device. Larvae were given ~2 min to get accustomed to their new environment before starting optogenetic experiments. Larval behaviour was recorded at room temperature for sequences of 2 min, including an initial 20–30 s long acclimation period. If not indicated otherwise, larvae were stimulated every 30 s.

The brightness and contrast of recorded movies were adjusted subsequently in ImageJ for better visibility. Larval length (Fig. 3e, f) was analysed in ImageJ using the last frame directly before light turn-on and 0.5 s after. The timing of forward and backward waves and when the muscle contraction wave was halted was manually tracked using the software Anvil[51]. From this, wave durations, the time until wave progression after light turn-off, and $t_{stop}$ were calculated. The spatiotemporal location of muscle contraction waves was tracked manually in ImageJ with the plugin MTrackJ. Off-time wave durations $t_{off}$ were calculated as the mean wave duration of up to two forward waves, each before and after stimulation. The stimulated segments were estimated visually from the movies.

Statistics, linear regression analysis with calculation of Pearson's r and significance tests via unpaired two-tailed t tests were performed in OriginPro.

## Data availability

The research data supporting this publication are available at https://doi.org/10.17630/7c9a6090-581b-474e-923d-320b2f9ce92c.

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

## Acknowledgements

We thank Dr. Maarten Zwart of the University of St Andrews for assistance with anatomical imaging experiments. This research was financially supported by the EPSRC NSF-CBET lead agency agreement (EP/R010595/1, 1706207), the DARPA-NESD programme (N66001-17-C-4012) and the Leverhulme Trust (RPG-2017-231). C.M. acknowledges funding from the European Commission through a Marie Skłodowska Curie individual fellowship (703387). S.R.P acknowledges funding from the Biology and Biotechnology Research council (BB/M021793) and from the Royal Society (RG150108). M.C.G. acknowledges funding from the Alexander von Humboldt Stiftung (Humboldt-Professorship). Diese Arbeit wurde mitfinanziert durch Steuermittel auf der Grundlage des vom Sächsischen Landtag beschlossenen Haushaltes.

## Author contributions

C.M. fabricated samples, performed measurements, analysed the data and prepared the paper. S.R.P. conducted anatomical experiments. S.R.P. and M.C.G. conceived and supervised the project. All authors contributed to the design of the study, interpretation of the data and preparation of the paper.

## Funding

## Competing interests

The authors declare no competing interests.
