## [Peer Review File · Nature Communications]

Reviewers' Comments:

Reviewer #1:

Remarks to the Author:

In this work, the research team demonstrated optogenetic control of behavior of *Drosophila melanogaster* larvae by illuminating their abdomen with micro-scale OLEDs, which resulted in stimulation of channelrhodopsins for either excitation or inhibition. They logically designed the experiment to show that stimulation in a specific segment for a certain time determines behavioral responses. They also analyzed and explained the test results in detail.

However, the reviewer does not think that the article provides enough novelty or scientific explanation to publish on Nature Communications. There seems no significant development of OLED devices, such as improved optical properties and transparency. In addition, there is a lack of identification of neurological mechanism, e.g., which certain neurons are responsible for the corresponding behaviors. At least, instead of neurological identification, which is too complicated, they should have presented a locally microscale neuron mapping, showing specific regions in segments are activated by sensory input.

For the above reasons, the reviewer recommends submitting this article to a more specialized journal.

Reviewer #2:

Remarks to the Author:

The work by Murawski, Pulver, and Gather describes a novel use of OLED technologies for optogenetics. While other light sources such as LEDs or waveguided light can also act as light sources for optogenetics, OLEDs can have a competitive edge over other technologies in that they can be made in the form of high-resolution arrays in which individual light sources can be independently modulated in space. By applying linear-array-type OLEDs as a means for the segmented light-induced stimulation of larvae, this manuscript illustrates exactly what this spatially-resolved OLED-based light sources can do for optogenetics. Reviewer believes the work presented by the authors is of high quality and will attract a significant amount of interest from both organic and bio communities; hence it should be published in Nature Communications after minor revisions addressing the following:

1. In Fig. 2 a or b, it would be nicer if authors provide a top-view schematic diagram as well, for the PDMS channel that hosts a larva.
2. In Fig. 2c, please clearly define the size of the scale bar shown at the bottom right corner.
3. It seems the spatial resolution of the proposed technique is ultimately limited by the scattering/optical diffusion that occurs within a larva itself. It would be nice if authors provide possible ways to improve the resolution of the present approach. In the same context, it would be even nicer if authors can provide information on a spatial resolution required by actual optogenetic applications.

Reviewer #3:

Remarks to the Author:

This manuscript describes the optogenetically controlled movement of *Drosophila* larvae using micro-structured OLEDs. The genetically modified *Drosophila* larvae with CsChrimson and GtACR2 showed different light-responsive behaviors under the light irradiation at whole body (A1~A8). While CsChrimson-expressing larvae showed full-body contraction, GtACR2-expressing larvae did the slow-down of forward wave or relaxation under the whole-body light irradiation using 4×4 mm² OLEDs. However, in the case of local stimulation, *Drosophila* larvae moved to the forward wave at posterior stimulation (A3-A8), but the backward wave at anterior stimulation (A1-A2).

This manuscript seems very interesting and acceptable after minor revision as commented below before publication.

Major Issues

[1] Authors should explain the reason for two different peak wavelengths of large-size OLEDs and micro-size OLEDs in Figure 1c.

[2] In page 6, although authors described the sensitivity of CsChrimson to 470 nm light from the intensity of 20 $\mu\text{W}/\text{mm}^2$, the experiment results showed that CsChrimson-expressed larvae could respond at the intensity of 2 $\mu\text{W}/\text{mm}^2$ in Figure 3c. In addition, authors described the fast GtACR2 response from the light intensity of 2 $\mu\text{W}/\text{mm}^2$, but GtACR2 showed lower response than CsChrimson in Figure 3c. Authors should explain the experiment data in Figure 3c more clearly using the light spectrum of the OLEDs.

[3] In Supporting Video 1, the brightness of OLED in two movies seems different, but the video caption describes the same value of 15 $\mu\text{W}/\text{mm}^2$. Authors should indicate the brightness of each video for Supporting Video 2 and 3, describing the ISO and f values for the video recording.

[4] In page 10, the heterozygous controls showed response at a high OLED intensity of 61 $\mu\text{W}/\text{mm}^2$. Authors should discuss the heat generation of OLEDs and compare the heat effect with that of visual input.

Minor Issues

[1] The terminology abbreviations should be fully described in the first appearance for easy understanding of readers in various fields. (e.g.) CS: Canton S and the meaning of ">" symbol. In addition, the optogenetic experimental methods should be described in more detail in the main manuscript or Supporting Information.

[2] It is recommended to cite the recent review article on optogenetic applications (Nature Reviews Materials 5, 149-165, 2020) for general readers in various fields.

Murawski et al, NCOMMS-20-10350

“Segment-Specific Optogenetic Stimulation in *Drosophila melanogaster* with Linear Arrays of Organic Light-Emitting Diodes”

Point-by-point reply

Our answers are written in blue.

Changes to the manuscript are given in orange.

Reviewer #1 (Remarks to the Author):

*In this work, the research team demonstrated optogenetic control of behavior of *Drosophila melanogaster* larvae by illuminating their abdomen with micro-scale OLEDs, which resulted in stimulation of channelrhodopsins for either excitation or inhibition. They logically designed the experiment to show that stimulation in a specific segment for a certain time determines behavioral responses. They also analyzed and explained the test results in detail.*

However, the reviewer does not think that the article provides enough novelty or scientific explanation to publish on Nature Communications. There seems no significant development of OLED devices, such as improved optical properties and transparency. In addition, there is a lack of identification of neurological mechanism, e.g., which certain neurons are responsible for the corresponding behaviors. At least, instead of neurological identification, which is too complicated, they should have presented a locally microscale neuron mapping, showing specific regions in segments are activated by sensory input.

For the above reasons, the reviewer recommends submitting this article to a more specialized journal.

We thank the reviewer for his/her thoughtful comments. To provide some context for the novelty of this work, we would like to clarify that our manuscript reports the first use of OLED technology to both inhibit and excite neural populations in a spatially restricted manner using optogenetic tools. We have achieved this by fabricating microstructured OLEDs that are developed and adapted specifically for this purpose and for use with larvae of *Drosophila melanogaster*. The main novelty in this work lies in the translation and adaptation of OLED technology for use by neuroscientists. The step from OLED development to actual use in biological preparations is non-trivial and often involves surprising challenges. We feel that there is real value in demonstrating how to undertake this step. The development of our controllable OLED array that is tailored for optogenetic experiments in a biological preparation paves the way for further work in a variety of model systems.

From a neurobiology perspective, our specific goal was to achieve large scale optogenetic activation and inhibition of sensory neurons in defined regions and to use this as a way to broadly raise and lower levels of excitation across the anterior-posterior (A-P) axis of the larval locomotor system. Controlling excitation levels across the A-P axis of motor systems has been postulated as a mechanism for initiation and coordination of motor programmes; however, carefully controlled experiments have been difficult to undertake in many systems. Our manuscript tests hypotheses based on this framework and thus it did not make sense to attempt to drill down to the level of single neurons as this would have reduced our ability to manipulate levels of excitation on network scale. Following this strategy to perform initial studies of sensory input to motor output relationships, we

were able to uncover several operating principles of the network, and this does provide new mechanistic insights into the operation of the system.

However, we agree that we could have been clearer about the approach taken. We have now added new text in the manuscript to further clarify our approach:

Introduction, page 4: The OLED light source developed here enables the projection of light onto specific areas of interest in a simple and reproducible manner and we use this to control sensory input along the anterior-posterior (A-P) axis of the animal. A long-standing hypothesis in motor systems research is that raising and lowering levels of excitability in motor circuits along the A-P axis represents a conserved mechanism for motor programme selection.^{18,19} In this study, we focused on segment-scale manipulation of peripheral sensory neuron activity using an optogenetic activator and inhibitor with the aim of controlling excitability along the A-P axis of the larval locomotor system. [...] Finally, we show that spatiotemporal patterns of sensory cell activation can entrain motor output depending on the activity state of the network.

Page 11: ... along the A-P axis of the animal.

Discussion, page 21: Overall, our work suggests that control of excitability along the A-P axis by sensory neurons in this system enables state-dependent initiation of motor activity and that motor networks within segments are tuned to be receptive to sensory input within defined windows relative to motor activity.

We also agree that further novel experiments would enrich the paper and further illustrate the novel capability offered by our method. To that end we have now added an additional figure (Fig. 7) and associated video (Supplementary Video 6) in which we deliver a moving wave of light across our preparation to show how the system can be used to deliver a spatiotemporal pattern of activation to peripheral sensory neurons. We analyse the dynamic motor-response of the larvae to these patterns in detail and discuss our results in the context of central pattern generating networks. This type of experiment has not been done to date, using OLEDs or other means of photo-stimulation, and we show that waves of sensory neuron activity can entrain motor output, but that the activity state of the network gates the influence of sensory input.

We have added the following Section to the manuscript (page 18-20):

Application of wave-like stimulation

The capability of our densely structured OLEDs to provide spatiotemporal stimulation is ideally suited to deliver complex light patterns. The muscle contraction waves that underlie forward crawling in *Drosophila* larvae are generated in the first instance by CPG networks in the larval central nervous system; however, the extent to which patterns of sensory neuron activity can entrain motor output has not been thoroughly examined.^{40,41} In the next set of experiments, we therefore investigated whether it is possible to influence and/or override the activity of CPG networks by delivering a wave-like sensory stimulation.

The typical wave duration of 5-40-GAL4 > CsChrimson larvae in our study was around 4 s (Fig. 5c). To mimic this crawling speed, we delivered a light wave across the abdomen from posterior to anterior, subsequently turning on six OLED pixels (each 100 μm in width) that were separated at a pitch of 400 μm for 0.5 s each. Thus, the light wave travelled a total distance of 2 mm over the course of 3 s. **Figure 1a** shows representative time-lapse images of larval response to the wave-like illumination sequence (Supporting Video 6). Larvae that were quiescent before illumination typically initiated a

muscle contraction wave that followed the wave-like stimulation, as evidenced by tracking the location of the muscle contraction waves of several larvae (Figure 1b). However, crawling patterns appeared somewhat less natural, showing tight contraction of illuminated segments and barely any contraction of thoracic segments (see Figure 1a and Supporting Video 6).

Figure 1. Response of 5-40-GAL4 > CsChrimson larvae to wave-like optical stimulation from posterior to anterior. a) Representative time-lapse images. The contracted larval segment is indicated by orange lines in each frame. Active OLED pixels are indicated in blue. Scale bar: 1 mm. See Supporting Video 6 for complete time-lapse. b) Overlay traces showing mean \pm SD of the location of muscle contraction waves head over time. Timing and location of the active OLED pixel is indicated by blue bars. $n = 3$ larvae, 9 stimulations. c) Wave durations grouped by start and end time of the wave relative to the optical stimulation. Before: wave ended before optical illumination. During: wave started before illumination and ended after illumination ceased. After: wave started after illumination. Starting at: wave started during illumination. $n = 4$ larvae. Two-tailed t-test: n.s. not significant ($p > 0.05$), * $p < 0.05$, ** $p < 0.01$, **** $p < 0.0001$. Whiskers: SD; diamond: mean; dashed line: median. d) Two representative traces showing the location of muscle contraction in response to slowed down wave-like stimulation. Light intensity: $30 \mu\text{W mm}^{-2}$.

Figure 1c compares the duration of muscle contraction waves before illumination, after illumination, for waves that started during illumination, and for waves that started before illumination and ended after illumination. Before illumination, wave durations were comparable to the previous results shown in Fig. 5c. Waves that were initiated by the light wave showed similar durations as before illumination, in line with the 3 s duration of the light wave. After illumination, waves significantly accelerated. Due to sensory activation, waves that started before illumination were significantly slowed down relative to all other conditions.

We also investigated the response of larvae to slower light waves (total wave duration of 6 s, with 6 steps of 1 s each). Two distinctive behavioural patterns were observed: If larvae were quiescent prior to stimulation, then they could entrain with the slow stimulation speed initially; however, the induced forward wave was converted into a backward wave when the illumination reached anterior abdominal segments (Figure 1d, top; Supporting Video 6). Larvae that were already crawling prior to

the stimulation could not adapt to the speed of the slow illumination and instead progressed with fast forward waves (Figure 1d, bottom; Supporting Video 6). This suggests that neural circuits within segments are tuned to receive sensory input within a specific time window during motor waves. Outside this temporal window, the network is less able to entrain with the speed of sensory stimulation.

Discussion Section on page 20/21: Additionally, the dense patterning of our OLEDs allowed us to deliver wave-like optical stimulation to sensory cells, with which we were able to entrain locomotor waves. While additional investigations are required to reveal the underlying mechanisms, our observations suggest the existence of a specific time window during motor waves during which motor circuits can be entrained by sensory input.

Supplementary Information, page 5: Video 6. Wave-like optical stimulation of 5-40-GAL4 > CsChrimson larvae from posterior to anterior by targeted local stimulation with a microstructured OLED. The location and periods during which the OLEDs were on are indicated in the video. OLED power density: $30 \mu\text{W mm}^{-2}$.

The reviewer suggests undertaking experiments aimed at ‘microscale mapping’ of synaptic partners downstream of sensory neurons. We cannot map on a neuron-by-neuron level which neurons downstream of sensory neurons are activated. (This would require live imaging of postsynaptic partners while simultaneously stimulating sensory neurons with OLEDs.) However, we have now performed additional confocal imaging measurements to provide anatomical expression patterns of the 5-40-GAL4 driver. This provides additional information on the expression pattern of our optogenetic tools and thus elucidates how our patterns of activation map to anatomical features in the larval nervous systems. This new data set is now included in the supporting information.

The following changes were made to the manuscript and supporting information:

Page 8: Analysis of the 5-40-GAL4 expression pattern confirmed expression in peripheral sensory neurons and in projections into the larval central nervous system (Supporting Fig. S2).

Methods Section, page 25: Anatomical analysis of expression patterns. 3rd instar 5-40 > CsChrimson animals were filleted and pinned in a sylgard lined dish using insect pins. Dissected preparations were fixed in 4 % paraformaldehyde, then washed 3-4 times in 0.1 M phosphate buffer. Preparations were mounted in Vectashield® (Vector Labs, Burlingame, CAe) and imaged with a Zeiss ApoTome.2 imaging system (Carl Zeiss Microscopy, Jena, Germany).

Supporting Figure S2:

Figure S2. Expression pattern of 5-40-GAL4. a) 3rd instar larval ventral nerve cord showing dense CsChrimson expression in neuropil regions (arrowhead) and nerve roots (arrows). Note lack of expression in any cell bodies in the central nervous system. b,c) 5-40-GAL4 expression in sensory neuron cell bodies in the peripheral nervous system (asterisks). Scale bars: 20 μ m in all panels.

Reviewer #2 (Remarks to the Author):

The work by Murawski, Pulver, and Gather describes a novel use of OLED technologies for optogenetics. While other light sources such as LEDs or waveguided light can also act as light sources for optogenetics, OLEDs can have a competitive edge over other technologies in that they can be made in the form of high-resolution arrays in which individual light sources can be independently modulated in space. By applying linear-array-type OLEDs as a means for the segmented light-induced stimulation of larvae, this manuscript illustrates exactly what this spatially-resolved OLED-based light sources can do for optogenetics. Reviewer believes the work presented by the authors is of high quality and will attract a significant amount of interest from both organic and bio communities; hence it should be published in Nature Communications after minor revisions addressing the following:

We thank the reviewer for his/her positive overall judgment of our work.

1. In Fig. 2 a or b, it would be nicer if authors provide a top-view schematic diagram as well, for the PDMS channel that hosts a larva.

We thank the reviewer for this suggestion and have now added a top-view schematic to Fig. 2(b):

Figure 2. ... Sketch ... b) of the larva mounted on top of the OLED and inside the PDMS channel (top view). ...

2. In Fig. 2c, please clearly define the size of the scale bar shown at the bottom right corner.

We have defined this now more clearly in the caption of Fig. 2: *Scale bars in c and d: 1 mm.*

We believe that Nature Communications style guides do not allow putting numbers directly into the figure but would be happy to adapt in line with editorial advice.

3. It seems the spatial resolution of the proposed technique is ultimately limited by the scattering/optical diffusion that occurs within a larva itself. It would be nice if authors provide possible ways to improve the resolution of the present approach. In the same context, it would be even nicer if authors can provide information on a spatial resolution required by actual optogenetic applications.

We thank the reviewer for this suggestion and have added some more information on how the resolution of our approach may be further improved to our discussion section on page 21/22: We expect that even smaller subsets of neurons in *Drosophila* larvae may be targeted by using red light to reduce scattering within larvae and by shaping the OLED emission more into forward direction, e.g., by increasing the strength of the optical microcavity.⁴³ Furthermore, flexible OLEDs could be shaped to form a tube through which larvae could crawl and which would give better optical access to neurons along the dorsal-ventral axis.

Furthermore, we added the spatial resolution requirements for our optogenetic application at the start of the results section on page 5: The device design and in particular the pixel size have to be adapted to the larval stage and the cells to be targeted. Here, we aimed at stimulating individual abdominal segments of third instar larvae, which are approximately 400 – 600 μm wide. We fabricated a...

Reviewer #3 (Remarks to the Author):

This manuscript describes the optogenetically controlled movement of Drosophila larvae using micro-structured OLEDs. The genetically modified Drosophila larvae with CsChrimson and GtACR2 showed different light-responsive behaviors under the light irradiation at whole body (A1~A8). While CsChrimson-expressing larvae showed full-body contraction, GtACR2-expressing larvae did the slow-down of forward wave or relaxation under the whole-body light irradiation using 4x4 mm² OLEDs. However, in the case of local stimulation, Drosophila larvae moved to the forward wave at posterior stimulation (A3-A8), but the backward wave at anterior stimulation (A1-A2). This manuscript seems very interesting and acceptable after minor revision as commented below before publication.

We thank the reviewer for his/her positive appraisal of our work.

Major Issues

[1] Authors should explain the reason for two different peak wavelengths of large-size OLEDs and micro-size OLEDs in Figure 1c.

We have now included an explanation on page 6/7 to explain the spectral differences between micro-sized and large OLEDs: The contributions from area and edge emission differ greatly between the microscopic OLED and the large OLED, with edge emission being substantially increased for the microstructured devices (c.f. the spread of light along the anode contact in Fig. 1b). This causes small deviations in the overall emission spectrum between both devices (Fig. 1c).

[2] In page 6, although authors described the sensitivity of CsChrimson to 470 nm light from the intensity of 20 $\mu\text{W}/\text{mm}^2$, the experiment results showed that CsChrimson-expressed larvae could

respond at the intensity of 2 $\mu\text{W}/\text{mm}^2$ in Figure 3c. In addition, authors described the fast GtACR2 response from the light intensity of 2 $\mu\text{W}/\text{mm}^2$, but GtACR2 showed lower response than CsChrimson in Figure 3c. Authors should explain the experiment data in Figure 3c more clearly using the light spectrum of the OLEDs.

Generally, the light intensity required depends on several conditions such as the specific driver lines used (Which neurons are targeted and where are they located? More superficial or deeper inside the animal?), developmental stage, wavelength and spectrum of the used light source, amount of all-trans retinal (ATR, the co-factor required for expression of the ChR) added to the food, and stimulation time. The light intensities mentioned on page 6 (20 $\mu\text{W}/\text{mm}^2$ for Chrimson and 2 $\mu\text{W}/\text{mm}^2$ for GtACR2) are taken from literature where the above experimental conditions will have been significantly different compared to conditions used in our experiments.

In literature, light intensity required to stimulate Chrimson-expressing *Drosophila* (20 $\mu\text{W}/\text{mm}^2$) was measured under the following conditions: Adult flies expressing Chrimson in gustatory neurons; Response measured by evaluating PER (proboscis extension reflex) scores; Flies raised on 0.2 mM ATR; Stimulation with 470 nm LED (providing rather narrowband spectrum and, thus, lower overlap with the Chrimson activation spectrum than for our rather broad-band OLED). [Klapoetke, N. C. et al. Nat. Methods 11, 338–46 (2014).]

The light intensity required to inhibit GtACR2-expressing *Drosophila* (2 $\mu\text{W}/\text{mm}^2$) was measured under the following conditions in the literature: Inhibition of glutamatergic neurons (vGlut-GAL4 driver) including motor neurons in larvae; 1 mM ATR; 457 nm LED. [Mauss, A. S., Busch, C. & Borst. Sci. Rep. 7, 13823 (2017).] Another publication reports an intensity of 14 $\mu\text{W}/\text{mm}^2$ for adult flies expressing GtACR2 in cholinergic neurons; 1 mM ATR; 460 nm LED. [Mohammad, F. et al. Nat. Methods 14, 271–274 (2017).]

Our experiments allow a direct comparison of light sensitivity between Chrimson and GtACR2 since we expressed both ChRs in the same subset of neurons and used the same illumination conditions. Under these specific and comparable conditions, we observed higher light-sensitivity for Chrimson-compared to GtACR2-expressing larvae.

In order to better explain our observations in the context of the current literature, we have now added the following paragraph on page 10: **In contrast to the intensity thresholds for activation reported in literature,^{25,31} in our experiments CsChrimson-expressing larvae generally showed a lower response threshold than GtACR2-expressing larvae. Note that the intensity requirements generally depend on the targeted cells, developmental stage, concentration of supplemented all-trans retinal (ATR), spectrum of the used light source, and stimulation time. Hence, a direct comparison to the literature is not possible.**

[3] In Supporting Video 1, the brightness of OLED in two movies seems different, but the video caption describes the same value of 15 $\mu\text{W}/\text{mm}^2$. Authors should indicate the brightness of each video for Supporting Video 2 and 3, describing the ISO and f values for the video recording.

A low intensity infrared LED was used for illumination when taking these videos. (Infrared was used to prevent undesired photostimulation from the light used for imaging.) The infrared light was delivered through a fibre bundle which had to be readjusted for each preparation. This readjustment led to considerable changes in the intensity of the infrared illumination between recordings. To compensate for these changes, the acquisition settings of the camera were then also adjusted. This in turn leads to an apparent change in the perceived brightness of the OLED pixels between recordings, even though the actual OLED intensity remained unchanged in Supporting Video 1.

In our revised manuscript, we have explained this now in more detail in the methods section (see our answer to Minor Issue [1] below). We mention the intensity level for each stimulation in Supporting Videos 2 and 3 directly in each of the videos (as described in the corresponding video caption).

[4] In page 10, the heterozygous controls showed response at a high OLED intensity of 61 $\mu\text{W}/\text{mm}^2$. Authors should discuss the heat generation of OLEDs and compare the heat effect with that of visual input.

While larvae can react to heat, temperatures between 18°C and 29°C are generally very well tolerated. Although the development time and growth rate of larvae vary significantly with temperature, no immediate effect on larval behaviour is expected from a brief temperature increase within this range.

A worst case estimate assuming that all electrical energy is converted into heat and not taking into account any heat dissipation by the water-filled PDMS sheet, gives a maximum temperature increase of our OLEDs of 3.4°C (after driving the device for 10 s at an intensity of 61 $\mu\text{W}/\text{mm}^2$). This worst case estimate is higher than measurements in literature, where temperature increases up to 1.1°C were observed when driving an OLED with much poorer performance (0.16 % EQE) at 30 V and 61 $\mu\text{W}/\text{mm}^2$ for 3 s.[B.F.E. Matarèse et al., Front. Bioeng. Biotechnol. 7 (2019) 278]. However, even a temperature increase of 3.4°C above ambient room temperature would not cause a behavioural response.

Furthermore, the observed response was immediate whereas any temperature increase will be more gradual and initially cumulative with illumination time.

We are thus confident that the response observed in controls is indeed due to photo-activation of body wall sensory neurons as explained in our manuscript.

To make this comparison clear to the reader, we have now added the following sentences to page 11: While OLEDs can heat up when driven at high currents for extended times, a worst case estimate (no heat dissipation, all electrical power converted to heat) shows that surface temperatures in our experiment will not have risen by more than 3.4°C; previous studies have measured a 1.1°C increase in temperature for similar light intensity levels but inferior OLED performance.¹⁴ Thus, it is highly unlikely that heating caused the observed response in control larvae. Instead, the response is likely due to activation of class IV body wall sensory neurons, which mediate a light avoidance behaviour upon exposure to blue light.³⁸

Minor Issues

[1] The terminology abbreviations should be fully described in the first appearance for easy understanding of readers in various fields. (e.g.) CS: Canton S and the meaning of ">" symbol. In addition, the optogenetic experimental methods should be described in more detail in the main manuscript or Supporting Information.

We now explain the meaning of the '>' symbol on page 8: We expressed UAS-GtACR2 and UAS-CsChrimson in all sensory neurons using the 5-40-GAL4 line,¹⁶ referred to hereafter as 5-40 > GtACR2 and 5-40 > CsChrimson, respectively. The '>' symbol denotes a driver (here, 5-40-GAL4) expressing a reporter that carries the transgene (here, GtACR2 or CsChrimson).

Additionally, we now also explain the meaning of CS on page 10: Heterozygous controls without CsChrimson or GtACR2 expression (i.e., crosses of 5-40-GAL4, CsChrimson, and GtACR2 to wild-type Canton Special (CS) flies)...

Furthermore, we now describe the optogenetic experimental methods in more detail (page 25/26): For experiments with wave-like illumination, OLEDs were addressed and driven by an Arduino Mega. The voltage output of the Arduino was converted to constant currents of 0.226 mA per pixel by constant current drivers (LM334Z). Videos were recorded as kinetic series in Andor Solis software at 25 Hz/16 bit and exported as tiff images. The exposure time was kept to 10 ms or less to avoid motion blur. Exposure time and camera gain were adjusted according to the brightness of the OLED and illuminating infrared LED.

For optogenetics experiments, feeding third instar larvae were taken out of the vials in dim light and gently washed in water. Then, larvae were slowly pushed into the water-filled PDMS channel from one side, while water was removed from the other side to suck the larvae into the channel. Subsequently, larvae were pinned down dorsal side up with an insect pin positioned between the trachea approximately in segment A5-A6. After fixing larvae in the PDMS channel, slight adjustment of the larval position with respect to the OLED pixel was possible by carefully sliding the PDMS sheet on the glass surface of the device. Larvae were given approximately 2 min to get accustomed to their new environment before starting optogenetic experiments. Larval behaviour was recorded at room temperature for sequences of 2 min including an initial 20-30 s long acclimation period. If not indicated otherwise, larvae were stimulated every 30 s.

Brightness and contrast of recorded videos were adjusted subsequently in ImageJ for better visibility.

[2] It is recommended to cite the recent review article on optogenetic applications (Nature Reviews Materials 5, 149-165, 2020) for general readers in various fields.

We thank the reviewer for pointing us to this recent publication which we have now added to the introduction as Ref. [4].

Reviewers' Comments:

Reviewer #1:

Remarks to the Author:

Through the revision of the manuscript, the authors well reflected the reviewers' comments. The authors claim that the excitation and inhibition of neural populations was first demonstrated by OLEDs as an optogenetic tool, which have advantages such as high resolution, flexibility and low toxicity. The reviewer agrees that this work has a novelty in the first application of OLEDs to neuroscientific research and will facilitate further research in other various models. Also, as the answer to the issue of lack of identification of neurological mechanism, the authors clarify that their goal is confined to large scale optogenetic stimulation of the larval locomotor system, not a neuron scale mapping, and give images of large scale expression patterns of the 5-40-GAL4 driver. Therefore, the reviewer believes, with several suggestions as below, this work should be published in Nature Communications.

[1] Although the authors answered other reviewers in page 6, the reviewer is just wondering if the difference in wavelength spectrum between micro OLEDs and large OLEDs is only due to the increased edge emission of micro OLEDs. The reviewer is wondering if there are other factors affecting the difference. It would be better to give explanation in more detail if possible.

[2] It is recommended to refer to the article which utilized inorganic micro LEDs for optogenetic stimulations. (Nano Energy 44 (2018) 447-455)

Reviewer #2:

Remarks to the Author:

Reviewer believes that authors have addressed all the issues raised in the previous review report and is ready for publication in Nat. Comm. Reviewer believes the addition of a new experimental result shown in Fig. 7 is also a good example illustrating the use of the proposed technology.

Reviewer #3:

Remarks to the Author:

Accept by Sei Kwang Hahn at POSTECH

Murawski et al, NCOMMS-20-10350A

“Segment-Specific Optogenetic Stimulation in *Drosophila melanogaster* with Linear Arrays of Organic Light-Emitting Diodes”

Point-by-point reply

Our answers are in blue.

Changes to the manuscript are marked orange.

Reviewer #1 (Remarks to the Author):

Through the revision of the manuscript, the authors well reflected the reviewers' comments. The authors claim that the excitation and inhibition of neural populations was first demonstrated by OLEDs as an optogenetic tool, which have advantages such as high resolution, flexibility and low toxicity. The reviewer agrees that this work has a novelty in the first application of OLEDs to neuroscientific research and will facilitate further research in other various models. Also, as the answer to the issue of lack of identification of neurological mechanism, the authors clarify that their goal is confined to large scale optogenetic stimulation of the larval locomotor system, not a neuron scale mapping, and give images of large scale expression patterns of the 5-40-GAL4 driver. Therefore, the reviewer believes, with several suggestions as below, this work should be published in Nature Communications.

We thank the reviewer for his/her positive judgment of our revised manuscript and for accepting our work.

[1] Although the authors answered other reviewers in page 6, the reviewer is just wondering if the difference in wavelength spectrum between micro OLEDs and large OLEDs is only due to the increased edge emission of micro OLEDs. The reviewer is wondering if there are other factors affecting the difference. It would be better to give explanation in more detail if possible.

In response to the question, we re-checked the processing conditions for micro OLED and large OLED and can confirm that both samples were fabricated in parallel, i.e., at the same time during the same thermal evaporation run. While this nominally ensures identical layer thickness, small thickness differences may still occur due to different positioning of the samples with respect to the rotation axis of the sample holder. (Samples sitting further away from the rotation axis will have slightly smaller layer thicknesses than samples sitting close to the axis.) Upon revisiting the spectra of all devices that were fabricated during this run, we found that the positioning of the samples indeed contributed to the observed differences in the spectrum. The figure below shows the spectra of the large OLED and the micro OLED at two different positions with respect to the rotation axis. Samples at position 1 were located further away from the rotation axis, which led to slightly smaller layer thicknesses and, thus, a spectral blue-shift. The devices used in our paper were the micro OLED from position 2 and the large OLED from position 1.

However, we also observe spectral differences between a micro OLED and a large OLED from the same position (i.e. same distance from the axis of rotation). Thus, we still believe that the edge emission effect plays a significant role. However, we also acknowledge that slight offsets in the layer thickness contributed as well to the spectral differences. While this does not change any conclusions made in our manuscript, we agree with the referee that it is best to discuss this point. Therefore, we have expanded our explanation on page 6/7 as follows:

The contributions from area and edge emission differ greatly between the microscopic OLED and the large OLED, with edge emission being substantially increased for the microstructured devices (c.f. the spread of light along the anode contact in Fig. 1b). Additionally, small differences in layer thickness are expected due to different positioning of the samples in the evaporation chamber. These effects caused small deviations in the overall emission spectrum between both devices (Fig. 1c).

[2] It is recommended to refer to the article which utilized inorganic micro LEDs for optogenetic stimulations. (*Nano Energy* 44 (2018) 447–455)

We thank the reviewer for pointing us to this publication which we have now added to the introduction as Ref. [10].

Reviewer #2 (Remarks to the Author):

Reviewer believes that authors have addressed all the issues raised in the previous review report and is ready for publication in Nat. Comm. Reviewer believes the addition of a new experimental result shown in Fig. 7 is also a good example illustrating the use of the proposed technology.

We thank the reviewer for his/her positive judgment of our revised manuscript.

Reviewer #3 (Remarks to the Author):

Accept by Sei Kwang Hahn at POSTECH

We thank Sei Kwang Hahn for his careful review and for accepting our revised manuscript.